# Detection of Circulating VZV-Glycoprotein E-Specific Antibodies by Chemiluminescent Immunoassay (CLIA) for Varicella–Zoster Diagnosis

**DOI:** 10.3390/pathogens11010066

**Published:** 2022-01-05

**Authors:** Arnaud John Kombe Kombe, Jiajia Xie, Ayesha Zahid, Huan Ma, Guangtao Xu, Yiyu Deng, Fleury Augustin Nsole Biteghe, Ahmed Mohammed, Zhao Dan, Yunru Yang, Chen Feng, Weihong Zeng, Ruixue Chang, Keyuan Zhu, Siping Zhang, Tengchuan Jin

**Affiliations:** 1Department of Dermatology, The First Affiliated Hospital of USTC, Division of Life Sciences and Medicine, University of Science and Technology of China, Hefei 230001, China; kombe@mail.ustc.edu.cn (A.J.K.K.); xiejiajia@ustc.edu.cn (J.X.); crx0419@163.com (R.C.); zhukeyuan1106@163.com (K.Z.); 2Hefei National Laboratory for Physical Sciences at Microscale, The CAS Key Laboratory of Innate Immunity and Chronic Disease, School of Basic Medical Sciences, Division of Life Sciences and Medicine, University of Science and Technology of China, Hefei 230026, China; ayesha1@mail.ustc.edu.cn (A.Z.); mahuan86@ustc.edu.cn (H.M.); taotao222@mail.ustc.edu.cn (G.X.); pb18081629@mail.ustc.edu.cn (Y.D.); amam@mail.ustc.edu.cn (A.M.); zhdan123@mail.ustc.edu.cn (Z.D.); yyr@mail.ustc.edu.cn (Y.Y.); cf155@mail.ustc.edu.cn (C.F.); whongz@ustc.edu.cn (W.Z.); 3Gabonese Scientific Research Consortium, Libreville, Gabon; Fleury.nsolebiteghe@cshs.org; 4Department of Radiation Oncology, Cedars Sinai Hospital, Los Angeles, CA 90048, USA; 5CAS Center for Excellence in Molecular Cell Science, Chinese Academy of Sciences, Shanghai 200031, China

**Keywords:** varicella–zoster virus (VZV), chemiluminescent immunoassay (CLIA), IgA, IgG, IgM, diagnostic test

## Abstract

Varicella and herpes zoster are mild symptoms-associated diseases caused by varicella–zoster virus (VZV). They often cause severe complications (disseminated zoster), leading to death when diagnoses and treatment are delayed. However, most commercial VZV diagnostic tests have low sensitivity, and the most sensitive tests are unevenly available worldwide. Here, we developed and validated a highly sensitive VZV diagnostic kit based on the chemiluminescent immunoassay (CLIA) approach. VZV-glycoprotein E (gE) was used to develop a CLIA diagnostic approach for detecting VZV-specific IgA, IgG, and IgM. The kit was tested with 62 blood samples from 29 VZV-patients classified by standard ELISA into true-positive and equivocal groups and 453 blood samples from VZV-negative individuals. The diagnostic accuracy of the CLIA kit was evaluated by receiver-operating characteristic (ROC) analysis. The relationships of immunoglobulin-isotype levels between the two groups and with patient age ranges were analyzed. Overall, the developed CLIA-based diagnostic kit demonstrated the detection of VZV-specific immunoglobulin titers depending on sample dilution. From the ELISA-based true-positive patient samples, the diagnostic approach showed sensitivities of 95.2%, 95.2%, and 97.6% and specificities of 98.0%, 100%, and 98.9% for the detection of VZV-gE-specific IgA, IgG, and IgM, respectively. Combining IgM to IgG and IgA detection improved diagnostic accuracy. Comparative analyses on diagnosing patients with equivocal results displaying very low immunoglobulin titers revealed that the CLIA-based diagnostic approach is overall more sensitive than ELISA. In the presence of typical VZV symptoms, CLIA-based detection of high titer of IgM and low titer of IgA/IgG suggested the equivocal patients experienced primary VZV infection. Furthermore, while no difference in IgA/IgG level was found regarding patient age, IgM level was significantly higher in young adults. The CLIA approach-based detection kit for diagnosing VZV-gE-specific IgA, IgG, and IgM is simple, suitable for high-throughput routine analysis situations, and provides enhanced specificity compared to ELISA.

## 1. Introduction

Varicella–zoster virus (VZV), also known as human herpesvirus-3 (HHV-3), belongs to the α-herpesviridae subfamily [1]. It is responsible for varicella disease or chickenpox in children, adolescents, and young adults. The latent viral resurgence years later commonly occurs in older people and causes a secondary infection known as zoster or shingles [1,2].

Although considered among the mild-symptom diseases, VZV-related diseases are highly morbid. The most life-threatening complications include mental development deficit, meningoencephalitis, and post-infectious encephalopathy, in varicella cases. In herpes zoster cases, complications include vasculitis, zoster sine herpete, and post-herpetic neuralgia. A lack of early diagnosis results in treatment delays, which usually leads to fatalities, especially in newborns, elders, organ transplant recipients, and immunocompromised people experiencing disseminated herpes zoster [3,4,5,6,7,8].

Nowadays, VZV vaccination has led to a significant decrease in the incidence of varicella, particularly in countries where vaccination programs have been implemented and well followed [9,10,11]. Consequently, this has decreased hospitalization and remarkably reduced the routine biological diagnoses in laboratories [12,13]. In these countries, VZV immunodiagnostic tests assessing IgA, IgG, and/or IgM have been only recommended in pregnant women, critically ill patients before organ transplant surgery, and immunocompromised people, post-vaccinated people, and hospital practitioners [14,15]. However, many reports hypothesized that the implementation of varicella vaccines would be followed by an increase in post-vaccine varicella or herpes zoster cases [2,9,10,16,17,18], suggesting the need for using antibody detection tests in routine diagnoses for global epidemiologic surveillance, along with herpes zoster vaccination [17]. For instance, in many other countries, such as China [18,19,20] and Norway [21], where anti-VZV vaccines are not yet implemented or where VZV vaccination coverage is uneven, rapid case identifications are crucial [19,21,22]. Reporting varicella cases in these countries, especially in high-frequented public areas such as schools, institutions, healthcare centers, hospitals, etc., would prevent rapid infection spread to people at risk (pregnant women, immunocompromised). Consequently, this will prevent them from progressing toward infection complication stages and facilitate outbreak control, as diagnosis delays are often fatal.

While the diagnosis of VZV infection is needed in both countries with well-established and non-implemented VZV-vaccine programs, routine biological diagnostics has become challenging, as many currently available diagnostic tests have low sensitivity/specificity [9,13,23]. The few highly sensitive immune diagnostic tests are scarce in the market or not evenly available worldwide [13]. Practically, in the past decade, several biological diagnostic tests with variable sensitivities have been developed to detect VZV-specific IgA, IgG, or IgM, or polyclonal antibodies. Most of them, including direct fluorescent antibody (DFA) and Tzanck smear diagnostic kits based on immunofluorescent assay (IFA), yield in low to moderate antibody detection sensitivity, around 60–80%, and 42–90%, respectively [24,25,26]. The VZV detection using virus culture assays resulted in high toxicities and contaminations, biasing the diagnostic results, as yielded mainly in false-negative (46% of sensitivity) [25]. In addition, these immunodiagnostic approaches are generally labor-intensive and time consuming, requiring meticulous specimen collection and highly trained technicians [25]. Those with high detection sensitivities and specificities (around 97.8% and 96.8%, respectively), including glycoprotein-based enzyme-linked immunosorbent assay (gpELISA) or varicella zoster glycoprotein IgG enzyme immunoassay with a reference time-resolved fluorescence immunoassay (VZV TRFIA), VZV IgG glycoprotein assay (Merck gpEIA) for the detection of serum VZV IgG, are not evenly and/or commercially available worldwide [27].

Although molecular diagnostics, including viral isolation from vesicular fluid cultures and swab samples, and nucleic acid detection (by PCR), are the most sensitive in VZV diagnostic [24,28], antibody assessment is needed in VZV epidemiological surveillance [11,12] and vaccination effectiveness control [9]. For instance, vesicular rashes do not always appear during infection, compromising PCR results [29,30]. Moreover, the main diagnostic approach based on clinical presentation is not 100% reliable, since many other herpetic diseases, including HSV, present similarly. Therefore, there is still a need for widely commercially distributed tests with high sensitivity and specificity for routine VZV diagnostics.

Several studies support that the chemiluminescent immunoassay (CLIA) approach for diagnosing serum/plasma viral antigen-specific IgA, IgG, or IgM has better diagnostic performance among the known immunoassays, including manual ELISA [31,32,33]. Manual ELISA is a labor-intensive multiple-wash-based assay; therefore, it is not suitable for high-throughput screening situations. Moreover, ELISA hardly detects weak antibody–antigen interactions and results in high background, compromising the test sensitivity. Interestingly, in the context of routine diagnostic of SARS-CoV-2, we and others [34,35] have developed CLIA-based diagnostic methods currently commercially available and with satisfying added values compared to ELISA [35] for diagnosing and monitoring COVID-19.

Therefore, we developed and validated a highly sensitive and specific diagnostic kit based on the CLIA approach for diagnosing VZV infections.

The developed CLIA-based VZV diagnostic approach demonstrated improved diagnostic accuracy, as it could detect very low IgA, IgG, and IgM titer in patients at the early stage of the VZV infection. Moreover, as the diagnosis process is automated, time-saving, and suitable for high-throughput situations, it can be used for routine diagnoses of VZV infections.

## 2. Results

### 2.1. Patient Characteristics and Sampling

Overall, 29 people (Appendix A) with an average age of 52 (20 to 82 years old) were enrolled and retained as VZV-infected patients, which was based on typical VZV symptoms and using ELISA tests. None of them was (or has been diagnosed) positive for herpes simplex viruses (HSV1/2), underwent an organ transplant surgery, or received an anti-VZV vaccine. Hepatitis A, B, and E were the only pathologies found among the patients. Appendix A describes the clinical and epidemiological characteristics of the included patients.

A total of 62 blood samples from the 29 retained VZV-patients and 453 plasmas/sera from random healthy people (used as negative controls) were obtained to test the developed diagnostic approach.

### 2.2. Patient Samples React with VZV Glycoprotein Depending on Concentration: Cohort Stratification

As aforementioned, three ELISA kits (ab108781, ab108782, and ab108783 for IgA/IgG/IgM; Abcam) were used to assess the VZV-specific IgA, IgG, and IgM, respectively, in VZV-patient samples from the 29 included VZV-patients. Sixteen samples from random healthy people were used as negative controls. According to the manufacturer’s instructions, these ELISA assays were considered standard for VZV-specific antibody detection in patients to confirm and stratify the retained patients as true-positive or equivocal groups. All samples were first serially diluted and assessed for VZV immunoglobulin detection. As a result, all patient samples (but not those from negative controls) reacted with VZV glycoproteins in a concentration-dependent manner (Appendix A). These results confirmed the presence of VZV-specific IgA, IgG, and IgM in VZV-patient samples.

Specifically, of the 29 included patients, 21 showed moderate to high reactivity, even at high dilution for IgA, IgG, and IgM detection, respectively. With OD450 ≥ 1 at 1/100 dilution, all these 21 patients were considered true-positive for IgA, IgG, and IgM according to the ELISAs’ manufacturer instructions (Figure 1, Appendix A).

In contrast, eight patient samples (patient number 6, 11, 13, 15, 18, 20, 21, and 24) showed inconsistent results in IgA, IgG, and IgM detection, respectively. Precisely, each patient sample showed equivocal results (OD450 between 0.1 and 0.21 at 1/100 dilution) for at least one of the three antibody isotypes IgA, IgG, and IgM (Figure 1, Appendix A). Therefore, they were considered as equivocal for further analyses.

### 2.3. Anti-VZV-gE IgA, IgG, and IgM Detection-Based VZV Diagnostic Kit Has High Sensitivity/Accuracy

The highly purified VZV-gE antigen (Appendix A) was used to make the CLIA-based IgG, IgA, and IgM detection kit, respectively. Then, this developed approach was tested for detecting VZV-gE-specific IgA, IgG, and IgM in sera and plasmas of patients. The reliability of the developed CLIA diagnostic kit was assessed with a two-fold serial dilution of the 62 samples from the 29 patients along with the 16 negative samples used in standard ELISA. As expecting and similar to the standard ELISA results, the CLIA approach showed sample dilution-dependent results (Appendix A). These results validated the CLIA diagnostic approach.

Then, to assess the performance of the CLIA approach, the cohort of 42 samples from the 21 true-positive patients and 453 independent samples (plasma or serum) from healthy people were tested by the developed VZV-gE-IgA/IgG/IgM kit. Before testing, samples were pre-treated (virus-inactivated) and diluted 20 times with dilution buffer (PBS) supplemented with 2% BSA. ROC analysis results showed sensitivities of 95.2% (IC95%: 76.2–99.9%), 95.2% (IC95%: 83.8–99.4%), and 97.6% (IC95%: 87.4–99.9%) and specificities of 98.0% (IC95%: 96.3–99.1%), 100% (IC95%: 99.2–100%), and 98.90% (IC95%: 97.4–99.6%) for IgA, IgG, and IgM detection, respectively (Figure 2A–C, Table 1). The cut-offs (criterion) for the IgA, IgG, and IgM diagnostic tests were >78662 RLU, >23450 RLU, and >89634 RLU, respectively (Figure 2A–C).

### 2.4. Combining IgM to IgG and IgA Detection Improves the Accuracy of the Varicella–Zoster Diagnosis

It is well known that IgMs are the first immunoglobulins produced at the first contact with viruses. Days after infection, IgM titer in the blood decreases while IgA and IgG titer increases [36]. As previously shown, adding IgA to serological CLIA improves the accuracy of SARS-CoV-2 diagnosis [34]. Then, we assessed whether the detection of IgM combined with that of IgG and/or IgA would be beneficial to the diagnosis of varicella and herpes zoster. As shown in Table 1, when VZV-gE specific IgM detection and one (but interestingly both) of the VZV-gE specific IgG and IgA detection were combined, the sensitivity, specificity, and the overall agreement increased significantly to 97.6%, 100%, and 99.8%, respectively. This combination has higher accuracy in diagnosing VZV than using the IgA, IgG, or IgM detection separately.

### 2.5. Equivocal Sample Analysis Shows Higher Sensitivity/Accuracy for CLIA Than ELISA in Combined Antibody Detection

To assess the diagnostic ability of the CLIA kit to diagnose the samples classified as equivocal by a standard ELISA tests, individual and combined detection of IgA, IgG, and IgM were conducted with the CLIA-based diagnostic approach in the eight-patient cohort. These results were analyzed along with previous cohort results and compared to those from ELISA. The comparative analysis was based on previously obtained IgA, IgG, and IgM criteria (>78662 RLU, >23450 RLU, and >89634 RLU, respectively) (Figure 2A–C) and the validation criteria of standard ELISAs.

For IgA detection, only patient samples 24 and 20 were diagnosed as positive by ELISA and CLIA, respectively (Figure 1, Table 2). No significant difference was observed in IgA levels between equivocal and negative controls (*p*-value = 0.567) (Figure 3A). Regarding IgG detection, all of these eight patients were determined equivocal, and none was positive by ELISA (Figure 1, Table 2). However, CLIA-based diagnostic revealed that two patient samples (patients 13 and one sample from 24) were positive (RLU > 23450), and the analysis of all the eight patient samples displayed RLU values significantly different from that of the negative controls (*p*-value < 0.001) (Table 2 and Figure 3B). Regarding IgM assessment, most of the ELISA-based equivocal samples were diagnosed as positive by CLIA (Figure 3C, Table 2). Overall, the combined diagnostic kits could detect lower immunoglobulin concentrations compared to ELISAs. These results suggest that CLIA-based diagnostic kits perform better than ELISAs in diagnosing varicella–zoster, especially in combination (Table 1 and Table 2).

### 2.6. Diagnostic Performance of CLIA-Based Immunoglobulin Diagnosis Regarding Equivocal Patients

We aimed to determine the diagnostic performance of the CLIA-based IgA, IgG, and IgM detection kit in diagnosing VZV infection in a random population. To this end, we considered the equivocal samples as positive and used the whole cohort, which consisted of the 62 independent samples from included patients and the 453 negative samples from healthy donors. The ROC analysis showed sensitivities of 74.2% (IC95%: 61.5–84.5%), 69.4% (IC95%; 56.3–80.4%), and 93.6% (IC95%: 84.3–98.2%), and specificities of 96.0% (IC95%: 93.8–97.6%), 99.8% (IC95%: 98.0–100%), and 98.0% (IC95%: 96.3–99.1%) for the diagnostic of VZV-gE IgA, IgG, and IgM, respectively (Figure 2D–F, Table 1). Interestingly, when the three VZV-gE IgA, IgG, and IgM detections were combined, the diagnostic performance was enhanced to sensitivity, specificity, and an overall agreement of 98.4%, 100%, and 99.8%, respectively (Table 1). Altogether, these analyses confirm that CLIA-based IgM detection alone or combined with IgA and IgG detection provides better diagnostic accuracy in diagnosing varicella and herpes zoster.

### 2.7. Antibody Titer Analysis in VZV Patients Suggests a Primary Infection in Equivocal Patients

Patient clinical data along with CLIA-based diagnostic results were analyzed. As a result, IgM titer was higher in all patients, while IgA and IgG titers were lower in patients with equivocal results (Figure 3 and Figure 4A). Moreover, since none of the included patients declared to have received VZV vaccine and with low IgA/IgG and high IgM titer, it was concluded that patients with equivocal results experienced varicella infection (primary infection). The high IgM detection associated with the presence of typical VZV-associated symptoms as early as two days in these patients (Appendix A) confirmed an acute phase of the varicella infection. However, unlike these eight patients, the 21 true-positive patients with high IgA, IgG, and IgM titers experienced an acute phase of viral reactivation (herpes zoster) or viral reinfection.

### 2.8. VZV-gE-Specific IgM Titer Negatively Correlates with Age

To assess whether there is any significant difference between the IgA, IgG, and IgM levels regarding VZV-patient age, the patient cohort was divided into two groups (≤35 years old and >35 years old). CLIA-associated diagnostic results from the 28 patients with reported age were considered (Appendix A). Analyses revealed that while no difference was observed in IgA/IgG titer, there was a significant difference in IgM level between the two groups. IgM level was significantly higher in young adults (below 35 years old) than in elder (Figure 4B).

## 3. Discussion

Reliable assays in the immunological diagnosis of VZV with the best performance are essential in the era of vaccine program implementation, their effectiveness evaluation, and for VZV-associated disease monitoring [12,14,18,19,20,23]. Most laboratories currently diagnose VZV-associated diseases from patient clinical symptoms, which is biased and may result in misdiagnoses identifying other (herpesvirus) infections with similar symptoms, such as HSV [25,37]. Moreover, VZV infections may usually present in atypical forms [38]. Various VZV immunodiagnostic tests developed so far lack sensitivities, and the most sensitive are unevenly available worldwide [9,13,23,24,25,26,27]. Here, we used a purified VZV-gE protein to develop and validate a highly-sensitive/accurate and automated CLIA approach for detecting IgA, IgG, and IgM specific to VZV in patient blood.

The developed CLIA diagnostic approach detected antibodies in VZV-patient samples with high accuracy/specificity and proportionally to the titer. Interestingly, this approach could detect very low antibody titers and determine positivity in most ELISA-based equivocal results. Moreover, the testing process was simple, fully automated, and thus suitable for high-throughput screening situations. The highly accurate results could be obtained as short as 50 min, with enhanced performance, making this CLIA-based diagnostic a better VZV-diagnostic tool than ELISA. The following Figure 5 describes the principle of CLIA approach as applied in Kaeser automate (Kangrun Biotech, Guangzhou, China).

The high accuracy demonstrated by the developed diagnostic kit in VZV-specific IgA, IgG, and IgM detection in blood samples was expected. Practically, infection with VZV induces robust antibody response, including IgA, IgG, and IgM antibodies produced mainly against VZV-gE and VZV-gI [1,39,40]. In fact, during the viral replication cycle, the VZV-gE is the most abundant glycoprotein produced and expressed on the VZV-infected cell surface, thus playing a central role in anti-VZV antibody production [1,39,40]. Moreover, while IgM is responsible for rapid and early immunity, long-term humoral immunity is initiated by the production of high-affinity IgG or IgA antibody. These circulating antibodies, especially the VZV-gE specific antibodies, are known to have the highest affinity to and neutralizing effect against VZV infection. During acute infections, IgA, IgG, and IgM antibody production is higher. However, in the absence of symptoms and in the earlier stage of infection, immunoglobulin production is low, and the most serum immunoglobulins produced are specifically directed toward VZV-gE, supporting our choice to use VZV-gE protein as serological antigen for developing this highly specific/accurate VZV diagnostic test [41,42]. Interestingly, Anna Grahn et al. [43] demonstrated that the use of VZV-gE in the detection of intrathecal specific antibodies is highly specific, without HSV non-specific reaction. Additionally, testing IgA together with IgG and IgM is crucial and has an added value in VZV diagnosis, because secretory IgA are mainly produced during VZV infection, as mucosal epithelial cells that mainly secrete IgA are the first cells to be infected with VZV.

The performance of some current commercially available VZV immunodiagnostic tests, such as VZV TRFIA and VaccZyme™ EIA, has been evaluated and reported [27]. For instance, from unvaccinated healthcare workers, VaccZyme™ EIA shows IgG detection sensitivities of up to 54.2% and specificities above 98.6%. On a comparable unvaccinated cohort, our developed CLIA-based VZV-gE IgG detection test showed better performance, with diagnostic sensitivity and specificity of 95.2% (IC95%: 83.8–99.4%) and 100% (IC95%: 99.2–100%), respectively excluding equivocal patients, and of 69.4% (IC95%: 56.3–80.4%) and 99.8% (IC95%; 98.0–100%), respectively considering equivocal patients (Table 1 and Figure 2).

There are a few available diagnostic kits that combine the simultaneous detection of VZV-IgA, IgG, and IgM. Moreover, the diagnostic performance of available IgM ELISA tests is lacking, especially in unvaccinated people [40,44,45]. Our CLIA-based detection kit, with high sensitivities and specificities in combined detection of IgA, IgG, and IgM specific to VZV in patient’s blood (Figure 2, Table 1 and Table 2), would be beneficial in routine diagnosis of varicella and herpes zoster. Furthermore, it can be of added value for post-vaccination immunity assessment, as good performance is expected in detecting low antibody levels commonly faced in vaccine recipients.

IgM is produced in high titer during the acute phase of primary infection [37,40], while IgG and IgA titers are low. Assessing IgA/IgG in this infection stage may result in false-negative-to-equivocal results. In the context of VZV infection diagnosis, such as in this study, such situations are usual [12] and require diagnosis confirmation 7 to 14 days later. However, in the presence of VZV infection symptoms such as rashes, the detection of VZV-specific IgM confirms the acute phase of the infection, although without specifying between primary, self-infection or reinfection, and viral reactivation. In the ELISA-based equivocal patients, the CLIA-based diagnostic approach showed low IgA/IgG and high IgM levels, suggesting that these patients (especially 6/8 patients) suffered from varicella in the acute phase. Pertinently, patients with equivocal results (except patients 18 and 21) visited the hospital as early as 2 to 3 days after the symptom onsets (Appendix A), thus supporting the hypothesis of an acute primary infection. For instance, studies of experimental simian varicella virus infection in monkeys demonstrated that IgG appears five days after IgM production, decreasing without competing with IgG [46]. A contrario, patients with a high level of IgA/IgG/IgM probably experienced herpes zoster.

The use of PCR in VZV diagnosis is preferred and recommended, as it is the most sensitive method to confirm varicella-zoster infection in vesicular lesions or scabs [12,24,28]. However, in the absence of rashes, this method is limited, with a decreased sensitivity, and results in false-negative when other samples, including blood, saliva, and cerebrospinal fluid (CSF), are used. Interestingly, it has been reported that in the absence of rashes, the use of blood and CSF to detect VZV-specific IgG by immunological methods yields more sensitivity than PCR for DNA detection [29,30]. Moreover, similar to other commercial immunological tests, PCR-based diagnostics are not widely available [12], and it is expensive and leads to patient compliance [47]. Altogether, our CLIA-based diagnostic tests filled these gaps and would be helpful in public health laboratories for routine varicella-zoster disease diagnosis, control outbreak situations [12], and varicella-zoster seroepidemiological studies for vaccine implementation purposes [14,18,19,20,23].

In this cohort, analysis of IgA, IgG, and IgM levels showed no significant antibody variation regarding gender (data not shown). However, regarding the age, it came out that while no difference in IgA and IgG level was found, the IgM level showed differences between young adults (below 35) and older. IgM tended to be higher in the youngest than in the eldest, which does not corroborate other studies, in which the antibody titers were proportional with age [20,48]. A larger population size would be preferred to draw a better relative conclusion. However, the conclusion on features and performance of our developed tests remains unaffected and valid.

However, although the population size permitted to validate the diagnostic approach, it was not large enough to better evaluate the performance of the test on a representative population scale, precisely to determine the predictive positive and negative values. For the same reason, evaluation of the correlation between each antibody and age range could have been biased as well. Therefore, future investigation with a large and representative population (including vaccinated and non-vaccinated) will better evaluate the performance of this diagnostic approach and study the immune response regarding age range. Moreover, it is suggested to use other samples, including saliva, which is thought to contain higher concentration of antibodies, specifically IgA in VZV infection, for a conclusive added value of IgA detection in the conventional immunodiagnostic kit.

In conclusion, detecting VZV-gE-specific IgA, IgG, and IgM using the developed kits based on the CLIA approach provided high sensitivity/accuracy and a rapid practical method for diagnosing VZV in unvaccinated individuals or determining VZV immune status after natural infection. This approach is simple, does not require outstanding trainees, and is suitable in high-throughput diagnosis situations.

## 4. Materials and Methods

### 4.1. Patient and Clinical Samples

This study was carried out under the approval (n° 2021-ky269) of the Medical Ethics Committee of the First Affiliated Hospital of the University of Science and Technology of China (USTC). From June to December 2020, a flow of patients was received in the hospital dermatological department for rashes, pimples, and other skin issues. Based on the presence of typical VZV symptoms (including paresthesia, localized pains, pimples, and non-oral and genital rashes), several patients were diagnosed as VZV-infected patients and included after obtaining free consent of participation. Two to three blood samples were collected into EDTA and dry tubes from each enrollee to investigate immunoglobulin (Ig) A, IgG, and IgM in plasma and serum, respectively. ELISA ab108781 (IgA), ab108782 (IgG), and ab108783 (IgM) tests (Abcam) were used as standards to exclude patient samples with negative results for all IgA, IgG, and IgM and retained those with at least one positive/equivocal result for IgA, IgG, or IgM, and thus stratified as true-positive or equivocal group. A total of 29 patients were retained, from which 62 blood samples were obtained for testing the developed CLIA diagnostic approach. Clinical patient data were obtained and listed in Appendix A.

Negative control samples were collected to assess the diagnostic accuracy. This cohort contained 453 samples from random healthy consenting people who did not report having suffered from or having been diagnosed positive for VZV infections and did not receive any VZV vaccines. All plasmas and sera were retrieved from EDTA (using Ficoll; density: 1.077) and dry tubes, respectively, by centrifugation. Retrieved plasmas/sera were treated with 1% TNBP and 1% Triton X-100 to completely denature any potential viruses [49] and stored at −20 °C (or −80 °C) until use.

### 4.2. Enzyme-Linked Immunosorbent Assay Tests

As aforementioned, three 96-well plate ELISA tests (Abcam ab108781, ab108782, and ab108783) were used to detect VZV-specific antibodies (IgA/IgG/IgM) in the enrolled patient samples as a complementary confirmation step. The ELISA tests were performed following the manufacturer’s instructions [15]. Before testing, samples were first virus-inactivated and then diluted accordingly with dilution buffer (PBS). For testing sera/plasmas, the manufacturer’s instructions were followed. The resulting yellow color intensity was measured at OD450 using a microplate reader. Each ELISA test was triplicated, and the data was graphed using GraphPad Prism 5 software.

Patient diagnostic results were determined from OD450 at dilution 1/100, as mentioned in the diagnostic kit leaflet. As a result of considerable background, the OD of the blank (PBS) was deducted from OD450 values of each sample result. Thus, a patient was considered positive when OD450 > 0.2, equivocal when OD450 was between 0.1 and 0.2, and negative when OD450 < 0.1. All patients negative for IgA, IgG, and IgM were systematically excluded from the study, while the retained patients were divided as true-positive or equivocal.

### 4.3. VZV Glycoprotein E Antigen Preparation

To develop the highly-sensitive diagnostic approach detecting VZV-specific IgA, IgG, and IgM in VZV-patient blood samples, the surface antigen glycoprotein E of VZV (VZV-gE) was produced from insect cell cultures using the baculovirus-based vector expression system (BVES) (Invitrogen-ThermoFisher Scientific, Waltham, MA, USA).

#### 4.3.1. Cell Cultures

Two insect cell lines, including *Spodoptera frugiperda* (*Sf9*) and *Trichoplusia ni* (*High Five*, *Hi5*), were used to produce the baculoviruses carrying the gene of interest and express gE protein, respectively. *Sf9* and *Hi5* were cultured at 27 °C in SIM-SF and SIM-HF medium (Sino Biological Inc., Beijing, China), respectively, and supplemented with 10% (*v*/*v*) heat-inactivated fetal bovine serum (FBS) and 1× penicillin/streptomycin. *Sf9* cell lines were maintained in both adherent and suspension cultures, while *Hi5* cell lines were only cultured in suspension culture. Adherent cell culture was carried out in 6 cm TC plates, and the suspension cultures were maintained in sterile autoclaved Erlenmeyer flasks in a 110 rpm spin shaker (27 °C). The suspension *Hi5* cell culture was diluted to 0.7 to 1 million cells every two based on cell viability and density. The adherents *Sf9* cell cultures were detached and diluted every three days based on cell viability and confluence. The cell viability was assessed under a fluorescence microscope using Trypan blue dye (0.4%) and counted using a hemocytometer.

#### 4.3.2. Molecular Cloning, Expression, and Purification of VZV-gE Protein

Briefly, the sequence of the mature extracellular region of VZV-gE (GenBank Accession number MH709377.1) was retrieved by PCR from the General Biosystem company’s synthetic construct using the following forward and reverse primers: 5′-ATTTCCAAGGTTCTTCCGTCTTGCGATACGATGATTTTCACATC-3′ and 5′-GACAAGCTTGGTACTTAATATCGTAGAAGTGGTGACGTTCCGGG-3′, respectively. In the meantime, the His-tag-modified transfer vector (pI-SUMO-Star-His) was linearized using primers (forward: 5′CTTCTACGATATTAAGTACCAAGCTTGTCGAGAAGTACTAGAGG3′ and reverse: 5′TATCGCAAGACGGAAGAACCTTGGAAATAAAGATTCTCGCTGCC3′) containing sequences that overlap the VZV-gE 5′ and 3′ end sequences. The linear fragments were ligated following the Gibson Assembly method’s instructions. The successful construct was transposed into bacmid using *DH10Bac E. coli* strain and purified for transfecting *Spodoptera frugiperda (Sf9)* insect cell lines, which produced recombinant VZV-gE baculoviruses. Expanded recombinant baculovirus stock was used to infect two million *Trichoplusia ni (High Five, Hi5)* insect cell lines for expressing the recombinant VZV-gE protein. The protein was harvested three days post-infection by high-speed centrifugation and purified from the supernatant.

A couple of purifications steps, including membrane diafiltration (Vivaflow 200), dialysis, ion-nickel column purification, size-exclusion chromatographic purification, and ultra-centrifugation were conducted to purify the protein. The purified protein (Appendix A) was stored in HEPES buffer saline (HBS: 20 mM HEPES, 250 mM NaCl), an amine-free buffer, which is required for the further experiments.

### 4.4. Preparation and Validation of the CLIA-Based Diagnostic Kit

The highly purified VZV-gE protein was employed to make the CLIA-based diagnostic kit. The purified VZV-gE protein was first biotinylated using EZ-Link Sulfo-NHS-LC-LC-Biotin, No-Weigh™ Format kit (Thermo Fisher, n°A35358, Waltham, MA, USA) following the manufacturer’s instructions. Then, the biotinylated protein was immobilized onto magnetic beads using an Invitrogen Dynabeads™ MyOne™ Streptavidin C1 kit (Thermo Fisher), following the manufacturer’s instructions, and further blocked (or saturated) with 2% of bovine serum albumin (BSA) to avoid non-specific interactions or background. Immobilizing the antigen protein onto a solid phase (here beads) is necessary for immunoblotting IgA, IgG, and IgM antibodies on a solid phase. The detection procedure below was performed with a fully automatic chemical luminescent immune analyzer, Kaeser 1000 (Kangrun Biotech, Guangzhou, China). Secondary antibodies anti-human IgA, IgG, or IgM conjugated with acridinium were used to detect the caught VZV-gE specific IgG, IgA, or IgM antibodies, respectively. The detected chemiluminescent signal over the background signal was automatically obtained as relative light units (RLU).

These collections, which contain all the described buffers and components for CLIA of VZV gE-specific IgA, IgG, and IgM, are referred to as VZV-gE-IgA, VZV-IgG, and VZV-gE-IgM kits here. As described above, each diagnostic kit was developed independently, with the corresponding secondary antibody conjugated with acridinium.

A first test batch of a two-fold serial dilution of the ELISA-based true-positive and healthy plasmas/sera was conducted to assess the reliability of the antibody detection kit regarding sample dilution. A subsequent CLIA test was performed to determine the diagnostic kit performances.

### 4.5. Statistical Analysis

ELISA tests were triplicate, and the results were transformed, fitted, and presented as mean ± SD. To determine the optimal cut-off values (criteria) and evaluate the diagnostic characteristics of VZV-gE-IgA, IgG, and IgM kits, receiver-operating characteristic (ROC) analyses were performed using MedCalc software. Thus, the specificity and sensitivity of the gE-specific IgA, IgG, and IgM detection kits were determined according to the following formulas:Sensitivity (%) = 100 × [True Positive/(True Positive + False Negative)];Specificity (%) = 100 × [True Negative/(True Negative + False Positive)];Overall agreement (%) = (True Positive + True Negative)/Total Tests.

A Mann–Whitney test was used to assess any significant variation of VZV gE-specific IgA, IgG, or IgM level between equivocal and true-positive ELISA-based categories. The same analysis was used to assess any significant correlation of the antibody levels regarding the age ranges. An analysis of variance (ANOVA) test was conducted using the Kruskal–Wallis approach to determine any difference of antibody level between the three independent groups, including positive, equivocal, and negative. A p-value less than 0.05 defined a hypothesis as statistically significant. All the above analyses were integrated into GraphPad Prism5.

## Figures and Tables

**Figure 1 pathogens-11-00066-f001:**
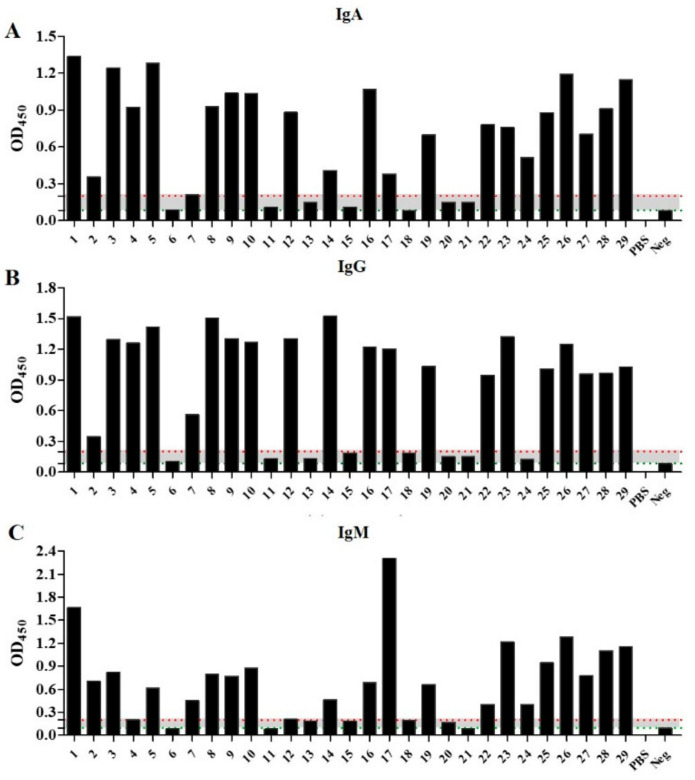
Results for confirmation and stratification of VZV infection in the recruited cohort. ELISA was performed using ab108781 (**A**), ab108782 (**B**), and ab108783 (**C**) ELISA tests, at 1/100 dilution for each sample. Patients with OD450 above 0.2 (red dotted line) were considered positive; with OD450 between 0.1 and 0.2 (the gray area between positive and negative threshold), they were equivocal; and with OD450 below 0.1 (green dotted line), they were considered negatives.

**Figure 2 pathogens-11-00066-f002:**
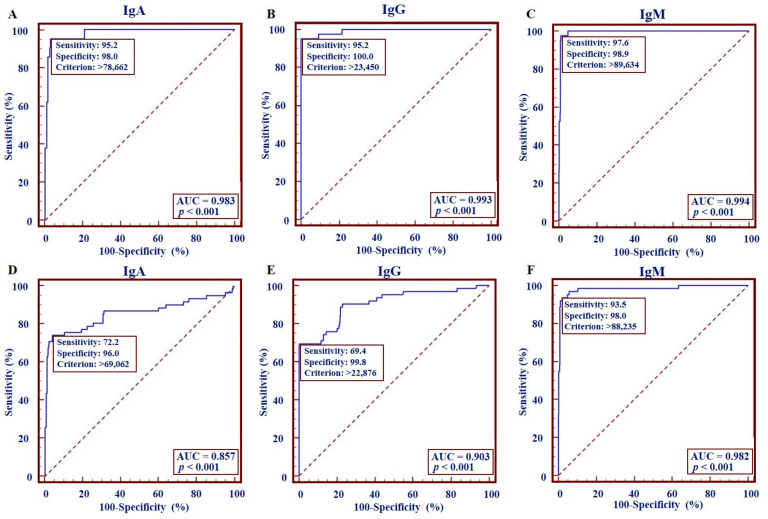
Performance of VZV-gE specific IgA, IgG, and IgM detection kits. The receiver-operating characteristic (ROC) curve analysis for detection of anti-IgA, IgG, and IgM antibodies against VZV-gE protein obtained from 42 ELISA positive samples, regardless (**A**–**C**), and considering as positive (**D**–**F**) the equivocal patient samples. The area under the curve (AUC) and the p-value are shown.

**Figure 3 pathogens-11-00066-f003:**
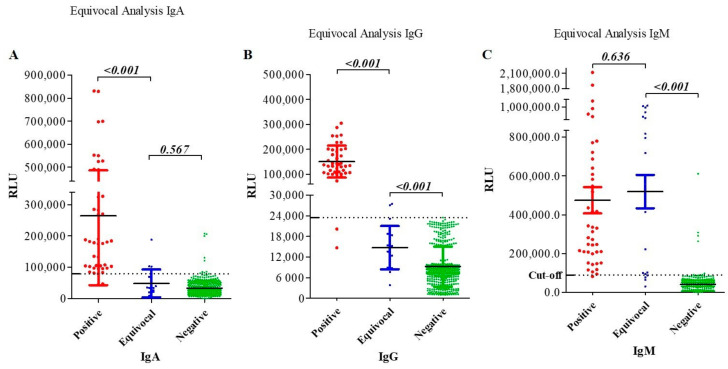
VZV-gE specific IgA, IgG, and IgM detection results and antibody levels in the patient cohort. Analysis of specific VZV serum antibody levels in highly positive (42 samples from 21 patients) and equivocal (20 samples from eight patients) patients revealed different levels in IgA (**A**), IgG (**B**), and IgM (**C**) antibody titers, as defined by the automated relative light units (RLU). The black bars in each distribution represent the mean, respectively, associated with the standards error of means (SEM). The dotted line indicates the cut-off values (>78,662 for IgA (**A**), >23,450 for IgG (**B**), and >89,634 for IgM (**B**)). RLU: relative light unit.

**Figure 4 pathogens-11-00066-f004:**
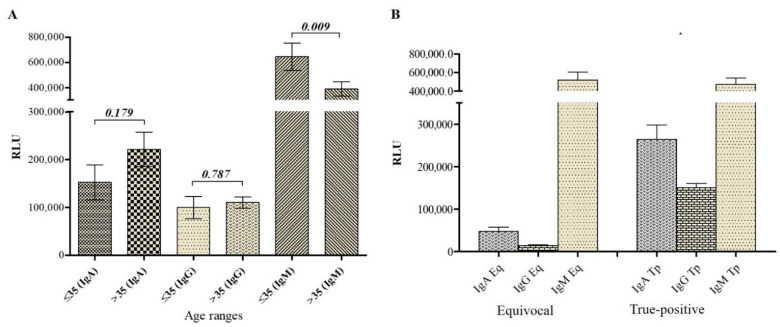
Antibody titer analysis in VZV patients. (**A**)**.** Analysis of CLIA-based diagnostic results demonstrated that patients with equivocal diagnoses were in the acute primary infection state. The high level of IgM in these patient samples corresponds to the early production of adaptative immunity (IgM), and the low titer of IgG corresponds to the gradient production of memory immunity. In contrast, patients with a high level of IgG and IgM were probably in the acute state of either reactivation or reinfection-associated herpes zoster. The high level of IgG and IgM demonstrate a simultaneous presence of active/acute memory immunity. Eq: equivocal; Tp: True-positive. (**B**). Variation of serum antibody level with age. Result analysis of IgA, IgG, and IgM antibody levels regarding the age range revealed a difference in IgM level (*p*-value < 0.05). RLU: relative light unit.

**Figure 5 pathogens-11-00066-f005:**
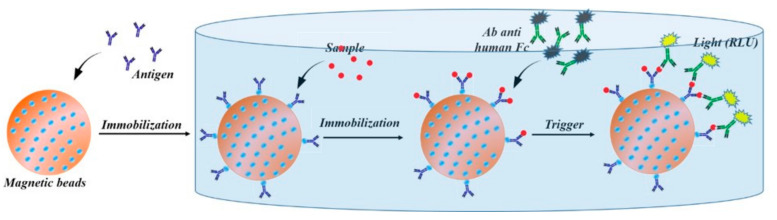
CLIA-based diagnostic assay principle. Purified VZV-gE antigen is immobilized onto metal beads and saturated with BSA. In the machine, a small amount of controls or test samples are added to the test tube and incubated. The test tube is washed to remove any unbound human immunoglobulin (h-Ig). A pre-labeled anti-human Ig conjugate is added to the test tubes. Then, a prepared substrate is added and catalyzed by the pre-labeled enzyme to produce a fluorescence, which is directly proportional to the amount of human anti-antigen Ig captured on the beads.

**Table 1 pathogens-11-00066-t001:** Sensitivity, specificity, and overall agreements of each VZV-gE-specific IgA, IgG, and IgM kit and their combinations in diagnosing varicella–zoster.

Antibody Type	Sensitivity			Specificity			Overall Agreement	
	n/Total	%	IC95%	n/Total	%	IC95%	n/Total	%
IgA *^#^*	40/42	95.2	76.2–99.9	444/453	98.0	96.3–99.1	484/495	97.8
IgG *^#^*	40/42	95.2	83.8–99.4	453/453	100	99.2–100	493/495	99.6
IgM *^#^*	41/42	97.6	87.4–99.9	448/453	98.9	97.4–99.6	489/495	98.8
IgA *^#^* & IgG *^#^*	38/42	90.5	NA	440/453	97.2	NA	480/495	97.0
IgG *^#^* & IgM *^#^*	40/42	95.2	NA	448/453	98.9	NA	488/495	98.6
IgA *^#^* & IgM *^#^*	40/42	95.2	NA	440/453	97.2	NA	480/495	97.0
IgA *^#^* & IgG *^#^* & IgM *^#^*	39/42	92.9	NA	440/453	97.2	NA	480/495	97.0
IgA *^#^* or IgG *^#^*	40/42	95.2	NA	453/453	100	NA	493/495	99.6
IgG *^#^* or IgM *^#^*	41/42	97.6	NA	453/453	100	NA	494/495	99.8
IgA *^#^* or IgM *^#^*	42/42	100	NA	452/453	99.8	NA	494/495	99.8
IgA *^#^* or IgG *^#^* or IgM *^#^*	41/42	97.6	NA	453/453	100	NA	494/495	99.8
IgA *	46/62	74.2	61.5–84.5	435/453	96.0	93.8–97.6	481/515	93.4
IgG *	43/62	69.4	56.3–80.4	452/453	99.8	98.0–100	495/515	96.1
IgM *	58/62	93.6	84.3–98.2	444/453	98.0	96.3–99.1	502/515	97.5
IgA * & IgG *	40/62	64.5	NA	434/453	95.8	NA	474/515	92.0
IgG * & IgM *	42/62	67.7	NA	443/453	97.8	NA	485/515	95.2
IgA * & IgM *	43/62	69.4	NA	427/453	94.3	NA	470/515	91.3
IgA * & IgG * & IgM *	39/62	62.9	NA	426/453	94.0	NA	465/515	90.3
IgA * or IgG *	49/62	79.0	NA	453/453	100	NA	502/515	97.5
IgG * or IgM *	59/62	95.2	NA	453/453	100	NA	512/515	99.4
IgA * or IgM *	61/62	98.4	NA	452/453	99.8	NA	513/515	99.6
IgA * or IgG * or IgM *	61/62	98.4	NA	453/453	100	NA	514/515	99.8

CLIA-based kit diagnoses features obtained regardless ^#^ and regarding as positive * the equivocal samples; NA: non-applicable.

**Table 2 pathogens-11-00066-t002:** VZV-gE-specific IgG, IgA, and IgM diagnostic results in the eight equivocal patient results.

Pat.	n°	RLU (OD_450_) Values			CLIA (ELISA) Results			Agreement Positivity (IgA or IgG or IgM)
		IgA	IgG	IgM	IgA	IgG	IgM	
6	1	34000 (0.099)	14883 (0.131)	795717 (0.09)	N (N)	N (E)	P (N)	P (E)
	2	19610 (0.076)	18775 (0.076)	932321 (0.076)	N (N)	N (N)	P (N)	
	3	14062 (0.089)	8502 (0.103)	1011533 (0.083)	N (N)	N (E)	P (N)	
11	1	2042 (0.136)	15398 (0.182)	717862 (0.073)	N (E)	N (E)	P (N)	P (E)
	2	8306 (0.089)	18254 (0.188)	816205 (0.088)	N (N)	N (E)	P (N)	
13	1	39269 (0.198)	23450 (0.159)	221901 (0.192)	N (E)	P (E)	P (E)	P (E)
	2	20987 (0.097)	47135 (0.097)	873343 (0.187)	N (N)	P (N)	P (E)	
	3	33650 (0.147)	27064 (0.129)	415041 (0.191)	N (E)	P (E)	P (N)	
15	1	69084 (0.157)	18366 (0.191)	995029 (0.169)	N (E)	N (E)	P (E)	P (E)
	2	34439 (0.069)	13626 (0.187)	854559 (0.188)	N (N)	N (E)	P (E)	
	3	56329 (0.11)	12408 (0.185)	1017070 (0.192)	N (E)	N (E)	P (E)	
18	1	34252 (0.071)	8940 (0.091)	64208 (0.111)	N (N)	N (N)	N (E)	N (E)
	2	26172 (0.091)	10172 (0.188)	78205 (0.191)	N (N)	N (E)	N (E)	
20	1	102290 (0.081)	7684 (0.198)	103009 (0.236)	P (N)	N (N)	P (P)	P (P)
	2	102454 (0.069)	8749 (0.099)	94009 (0.068)	P (N)	N (E)	P (N)	
	3	187560 (0.146)	11790 (0.151)	100154 0.197)	P (E)	N (E)	P (E)	
21	1	25117 (0.198)	3836 (0.075)	30841 (0.062)	N (E)	N (N)	N (N)	N (E)
	2	78099 (0.084)	15208 (0.153)	88775 (0.074)	N (N)	N (E)	N (N)	
24	1	3365 (0.513)	27405 (0.189)	571206 (0.401)	N (P)	P (E)	P (P)	P (P)
	2	69084 (0.601)	13048 (0.089)	605542 (0.285)	N (P)	N (N)	P (P)	
Total		-	-		3(2)/20	4(1)/20	16(3)/20	6(2)/8

E: Equivocal; P: Positive; N: Negative; RLU: Relative Light Unit. The number of samples for each patient is determined by grey difference.

## Data Availability

Restrictions apply to the availability of the patients data. Patient data was obtained from the First Affiliated Hospital of the University of Science and Technology of China (USTC) and are available from the corresponding authors with the permission of the First Affiliated Hospital of the University of Science and Technology of China (USTC), while patient sample-related experimental research data was obtained from our Laboratory.

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
