# Peer review of "Detection of Circulating VZV-Glycoprotein E-Specific Antibodies by Chemiluminescent Immunoassay (CLIA) for Varicella–Zoster Diagnosis"

_pathogens, 2022, doi:10.3390/pathogens11010066_

Round 1

Reviewer 1 Report

This manuscript by Kombe, et al. is well written and presented. It is an interesting and straightforward study. I think the results are of benefit to the scientific community and may help to shape future methods for VZV diagnostic testing.

My only suggestion for improvement would be to better define and explain the patient cohorts in the body of the paper (e.g., Results section 2.1 or Materials section 4.1).

For the VZV-infected patients, were these all confirmed to be primary infections? What was the average length of time between diagnosis and sample acquisition? How many were confirmed VZV positive by PCR?

Additionally, for the negative control samples, how many had previously received the VZV vaccine? Or were they recruited because they reported no known prior VZV infection and confirmed to never have been vaccinated?

Answers to these questions would make the paper more clear and easier to understand.

Author Response

Response to Reviewer 1 Comments

[General Comment]. This manuscript by Kombe, et al. is well written and presented. It is an interesting and straightforward study. I think the results are of benefit to the scientific community and may help to shape future methods for VZV diagnostic testing.

[Response] Thank you very much.

[Comment 1] My only suggestion for improvement would be to better define and explain the patient cohorts in the body of the paper (e.g., Results section 2.1 or Materials section 4.1).

[Response 1] We thank the reviewer for this suggestion. We have defined the cohort patients in the revised version of the manuscript, specifically in Materials section 4.1, through the new figure depicting the flow chart of patient inclusion and analyses, add in supplementary materials, as “Supplementary Figure S1”.

[Comment 2] For the VZV-infected patients, were these all confirmed to be primary infections? What was the average length of time between diagnosis and sample acquisition? How many were confirmed VZV positive by PCR?

[Response 2] We thank the reviewer for pointing this comment out. Not all the VZV-infected patients were experiencing primary infection. Based on our analysis results, only the 8 VZV-patients with ELISA-based equivocal results were concluded through CLIA test to experience the primary infection because they showed high IgM but, low IgG/IgA levels, along with the presence of VZV-typical symptoms marking an acute infection state.

The average length of time between the sample acquisition and diagnosis was around one week after collecting the last patient samples.

PCR was not used in this study to confirm VZV infection in included patients, because we could only collect blood samples from all patients. And, it is well known that assessing VZV-nucleic acid using blood samples is not recommended as it is associated with high rate of false-negative results. Unlikely, ELISA is highly recommended when blood is used as sample for diagnosing VZV.

[Comment 3] Additionally, for the negative control samples, how many had previously received the VZV vaccine? Or were they recruited because they reported no known prior VZV infection and confirmed to never have been vaccinated?

[Response 2] We thank the reviewer for pointing this comment out. The negative control samples were obtained from healthy people who reported no known recent prior VZV infection and confirmed to have never been vaccinated with VZV vaccines.

Reviewer 2 Report

Interesting paper on highly sensitive and specific chemiluminescent immunoassay for circulating VZV-specific antibodies detection. The paper needs however major revisions.

Title: The current title is not sufficiently informative. I would suggest a title such as “Detection of circulating VZV-glycoprotein E-specific antibodies by chemiluminescent immunoassay” or “Detection of circulating VZV-glycoprotein E-specific antibodies by chemiluminescent immunoassay for varicella-zoster diagnosis”.

Abstract:

Line 25; add: (gE) after VZV-glycoprotein E

Lines 26, 34: Write: IgA, IgG and IgM, and not IgG, IgA and IgM; overall: present in the same order IgA, IgG and IgM

Line 35; proved (and not proves)

Introduction:

Line 56: Delete “which is difficult to treat”

Line 63: English unclear; please correct

Line 68: Delete “huge”

Line 70: in (not In)

Line 82: English unclear (“since”); please correct

Line 103: Delete “urgent”

Line 118; Please write: IgA, IgG and IgM

Line 136: English unclear; please correct

Results

Overall (text and figures): present in the same order IgA, IgG and IgM results

Line 195: Delete “2”

Table 1. Write overall aa.a% and not aa.aa%: n/total before %; add IC95%; overall in Results section and text

Table 2. Define RLU; note RUL and not RUL

Discussion

Line 273 (overall). Please explain and discuss the interest of VZV-specific IgA, IgG and IgM detection; why IgA ?

Line 273 (overall). Please explain how the specificity of VZV-specific IgM was assessed

Line 283: Please give the acronym CLIA only once in the text

Overall: Discuss the strength and weakness (weak size of included patients) of the study

Material and Methods

Line 351 (paragraph): Please add a flow chart for inclusion, and analyses

Author Response

Response to Reviewer 2 Comments

[General Comment]. Interesting paper on highly sensitive and specific chemiluminescent immunoassay for circulating VZV-specific antibodies detection. The paper needs however major revisions.

[Response] Thank you very much. We have read and considered each and all your comments and tried to address them one by one to provide the best revision in order to fit your standard

Title

[Comment 1]: The current title is not sufficiently informative. I would suggest a title such as “Detection of circulating VZV-glycoprotein E-specific antibodies by chemiluminescent immunoassay” or “Detection of circulating VZV-glycoprotein E-specific antibodies by chemiluminescent immunoassay for varicella-zoster diagnosis”.

[Response 1] We thank the reviewer for this suggestion. We agreed to change the title as follows: “Detection of circulating VZV-glycoprotein E-specific antibodies by chemiluminescent immunoassay (CLIA) for varicella-zoster diagnosis”.

Abstract

[Comment 2] Line 25; add: (gE) after VZV-glycoprotein E.

[Response 2] We thank the reviewer for this suggestion. We modified accordingly.

[Comment 3] Lines 26, 34: Write: IgA, IgG and IgM, and not IgG, IgA and IgM; overall: present in the same order IgA, IgG and IgM.

[Response 3] We thank the reviewer for pointing these out. We addressed accordingly in the whole revised manuscript.

[Comment 4] Line 35; proved (and not proves)

[Response 4] We thank the reviewer for pointing these out. We addressed accordingly.

Introduction:

[Comment 5] Line 56: Delete “which is difficult to treat”

[Response 5] We thank the reviewer for this suggestion. We addressed accordingly.

[Comment 6] Line 63: English unclear; please correct

[Response 6] We thank the reviewer for this comment. We agreed and modified accordingly in the revised manuscript.

[Comment 7] Line 68: Delete “huge”

[Response 7] We thank the reviewer for this suggestion. We modified accordingly.

[Comment 8] Line 70: in (not In)

[Response 8] We thank the reviewer for this comment. We agreed and modified accordingly in the revised manuscript.

[Comment 9] Line 82: English unclear (“since”); please correct

[Response 9] We thank the reviewer for this suggestion. We agreed and modified accordingly in the revised manuscript.

[Comment 10] Line 103: Delete “urgent”

[Response 10] We thank the reviewer for this suggestion. We agreed and modified accordingly in the revised manuscript.

[Comment 11] Line 118; Please write: IgA, IgG and IgM

[Response 11] We thank the reviewer for the comment and suggestion. We agreed and modified accordingly in the revised manuscript.

[Comment 11] Line 118; Please write: IgA, IgG and IgM

[Response 11] We thank the reviewer for the comment and suggestion. We agreed and modified accordingly in the revised manuscript.

[Comment 12] Line 136: English unclear; please correct

[Response 12] We thank the reviewer for the comment and suggestion. We agreed and modified the unclear English in the revised manuscript.

Results

[Comment 13] Overall (text and figures): present in the same order IgA, IgG and IgM results

[Response 13] We thank the reviewer for the constructive comment and suggestion. We modified and presented as IgA, IgG, and IgM like suggested in the entire revised manuscript.

[Comment 14] 195: Delete “2”

[Response 14] We thank the reviewer for the comment. We modified accordingly in the revised manuscript.

[Comment 15] Table 1. Write overall aa.a% and not aa.aa%: n/total before %; add IC95%; overall in Results section and text

[Response 15] We thank the reviewer for this comment. We modified accordingly in the revised manuscript.

[Comment 16] Table 2. Define RLU; note RUL and not RUL

[Response 16] We thank the reviewer for the comment and suggestion. We defined accordingly in the revised manuscript.

Discussion

[Comment 17] Line 273 (overall). Please explain and discuss the interest of VZV-specific IgA, IgG and IgM detection; why IgA?

[Response 17] We thank the reviewer for the comment and suggestion. We justify the interest of IgA, IgG, and IgM in the discussion section.

[Comment 18] Line 273 (overall). Please explain how the specificity of VZV-specific IgM was assessed.

[Response 18] We thank the reviewer for this comment. We addressed accordingly.

[Comment 19] Line 283: Please give the acronym CLIA only once in the text.

[Response 19] We thank the reviewer for the comment and suggestion. We modified accordingly in the revised manuscript.

[Comment 20] Overall: Discuss the strength and weakness (weak size of included patients) of the study.

[Response 20] We thank the reviewer for the constructive suggestion. We addressed accordingly in the revised manuscript.

Material and Methods

[Comment 20] Line 351 (paragraph): Please add a flow chart for inclusion, and analyses.

[Response 20] We thank the reviewer for the constructive suggestion. We agree, and we have added a new figure depicting the flow chart for patient inclusion and analyses, in supplementary materials, as “Supplementary Figure S1”.

Reviewer 3 Report

The manuscript of Arnaud John Kombe Kombe et al. describes the development of a chemiluminescent immunoassay for the detection of the varicella-zoster virus-specific IgG, IgA and IgM.

In my opinion the manuscript is well written and well organized and would be of interest for the Pathogens readership.

Below some suggestions to improve the manuscript.

In the introduction section.

Page 3 lines 108-110. Since the CLIA usually suffers the same issues of the ELISA I suggest to revise the statement. Moreover, I remind the authors that also the ELISA can be automatized.

In the results section.

Page 3 line144. Please revise the numeration: supplementary figure s2 should be s1.

Page 4 line 177. Please add the unit of measurement (the same for line 204).

Page 8 lines 260-262. Please remove

In the Materials and Methods section.

Page 10 lines 375,376. “The ELISA 96-well plates were prepared as previously described [15].” Are you referring to the commercial ELISA tests? If yes I did not understand why you reported the abovementioned sentence.

Page 11 line 378 “fluorescence”. Actually, I think that the ELISA tests used are not based on fluorescence signal.

Page 12 line 426. Check “puify the protei”

Additional comments.

A scheme regarding the working principle of the test should be added.

I suggest to add the very brief conclusions at the end of the Discussion section and to remove the conclusions section.

Author Response

Response to Reviewer 3 Comments

[General Comment]. The manuscript of Arnaud John Kombe Kombe et al. describes the development of a chemiluminescent immunoassay for the detection of the varicella-zoster virus-specific IgG, IgA and IgM.

In my opinion, the manuscript is well written and well organized and would be of interest for the Pathogens readership.

Below some suggestions to improve the manuscript.

[Response] Thank you very much. We have read and considered all your comments and tried to address them one by one to provide the best revision in order to fit your standard.

In the introduction section.

[Comment 1]: Page 3 lines 108-110. Since the CLIA usually suffers the same issues of the ELISA I suggest to revise the statement. Moreover, I remind the authors that also the ELISA can be automatized.

[Response 1] We thank the reviewer for pointing this out. In that statement, we meant ELISA as a manual assay, which has lower performances than CLIA. Moreover, even though ELISA can be automatized, there is no automatized ELISA kit for VZV, and the automatized ELISA is better than the manual assay.

However, we revised the statement.

In the results section

[Comment 2] Page 3 line144. Please revise the numeration: Supplementary Figure S2 should be S1.

[Response 2] We thank the reviewer for pointing this out. We re-organized the figures in supplementary materials accordingly and revised the numeration.

[Comment 3] Page 4 line 177. Please add the unit of measurement (the same for line 204).

[Response 3] We thank the reviewer for pointing this out. We addressed accordingly in the revised manuscript.

[Comment 4] Page 8 lines 260-262. Please remove

[Response 4] We thank the reviewer for pointing this out. We removed the two sentences.

In the Materials and Methods section

[Comment 5] Page 10 lines 375,376. “The ELISA 96-well plates were prepared as previously described [15].” Are you referring to the commercial ELISA tests? If yes I did not understand why you reported the abovementioned sentence.

[Response 5] We thank the reviewer for pointing this out. Indeed, we are referring to the commercial tests. We followed the design and protocol of the commercial ELISA kits so that the result would be more comparable. We revised accordingly.

[Comment 6] Page 11 line 378 “fluorescence”. Actually, I think that the ELISA tests used are not based on fluorescence signal

[Response 6] We thank the reviewer for pointing this out. We agree, and we modified as “yellow color intensity”.

[Comment 7] Page 12 line 426. Check “puify the protei”

[Response 7] We thank the reviewer for this suggestion. We modified accordingly.

Additional comments

[Comment 8] A scheme regarding the working principle of the test should be added.

[Response 8] We thank the reviewer for this constructive suggestion. We agree, and we have added a new figure describing the working principle, in discussion section as “Figure 5”.

[Comment 9] I suggest to add the very brief conclusions at the end of the Discussion section and to remove the conclusions section.

[Response 9] We thank the reviewer for this constructive suggestion. We have modified accordingly.

Round 2

Reviewer 2 Report

The authors have precisely addressed all issues raised by the first reviewer, and the paper may now be considered desirable for publication.

Reviewer 3 Report

The authors have addressed all the reviewers' concerns by improving the manuscript. I suggest the manuscript publication.